# Prevalence of Precancerous Cervical Lesions among Nonvaccinated Kazakhstani Women: The National Tertiary Care Hospital Screening Data (2018)

**DOI:** 10.3390/healthcare11020235

**Published:** 2023-01-12

**Authors:** Balkenzhe Imankulova, Aisha Babi, Torgyn Issa, Zhanar Zhumakanova, Ljubov Knaub, Aidana Yerzhankyzy, Gulzhanat Aimagambetova

**Affiliations:** 1Clinical Academic Department of Women’s Health, CF University Medical Center, Astana 010000, Kazakhstan; 2Department of Biomedical Sciences, School of Medicine, Nazarbayev University, Astana 010000, Kazakhstan; 3Clinical Academic Department of Laboratory Medicine, Pathology and Genetics, CF University Medical Center, Astana 010000, Kazakhstan

**Keywords:** cervical cancer, precancerous cervical lesions, CIN, HSIL, LSIL

## Abstract

Objective: At the present time, cervical cancer remains the fourth most prevalent cancer among women worldwide. Most cervical cancer cases are attributed to high-risk human papillomavirus (HPV) infection. Because the natural history of cervical cancer takes decades, the disease could be prevented if premalignant conditions are identified and appropriately managed. The aim of this study is to identify the prevalence of precancerous lesions among non-vaccinated women attending the national tertiary care hospital in Kazakhstan. Methods: This was a retrospective study of the cervical cancer screening database (2018) from the national tertiary care hospital in Kazakhstan. Records of 6682 patients, who had cervical cytology tests by Papanicolaou (Pap test), were analyzed. Out of the revised cases, 249 patients had abnormal cervical cytology reports. The Pap test was performed using liquid-based cytology (LBC). The data were analyzed using the statistical software STATA 16. A *p*-value of less than 0.05 was considered statistically significant. Results: In this retrospective analysis of 6682 patients’ records, we found 3.73% (249 patients) out of all Pap tests performed in 2018 were abnormal. The prevalence of high-grade squamous intraepithelial lesion (HSIL) was high at 19.28%, and the proportion of atypical squamous cells of undetermined significance (ASCUS) and atypical squamous cells (ASCs-H) was 18.47%, while low-grade squamous intraepithelial lesion (LSIL) were identified in 62.25% of the cases. Almost 25% of the women included in the study had concurrent lower and upper genital tract infections. Conclusion: Although the overall rate of abnormal Pap test results was not high, the study shows the elevated prevalence of HSIL. It calls the attention of local policymakers and gynecology specialists and requires immediate actions to improve the prophylactic measures to decrease morbidity and mortality from cervical cancer in Kazakhstan.

## 1. Introduction

At the present time, cervical cancer remains the fourth most prevalent cancer among women worldwide [1,2,3,4]. Moreover, most cervical cancer cases (85%) occur in low- and middle-income countries (LMICs), where cervical cancer remains the leading cause of cancer-related mortality [2]. Each year, the number of cervical cancer cases dynamically increases globally [3,5]. Between 2000 and 2018, the absolute number of cervical cancer cases increased from 471,000 to 570,000 [2,6]. In 2018, there were around 311,000 deaths from cervical cancer [2]. Moreover, by 2030, it is estimated that the number of new cervical cancer cases will reach 700,000 and the annual number of deaths will reach 400,000 [7]. Therefore, given such a significant annual increase in the number of cases and deaths, the elimination of cervical cancer is a global public health challenge [8,9].

In most cases, cervical cancer is caused by high-risk human papillomavirus (HPV), and 16 and 18 HPV types are responsible for 70% of cervical cancers and precancerous cervical lesions [7]. HPV is one of the most common sexually transmitted infections in the world [7]. The prevalence of HPV varies between countries globally from 7 to 14% in Australia and Middle Eastern countries to the highest prevalence amid women in the developing countries of Eastern and Southern Asia, which is reported from 36.3% to 44.4% [10,11].

Because high-risk HPV types are known to be the main causative agents for cervical cancer, the primary prevention of related malignant and premalignant conditions largely depends on HPV vaccination [8,9,12,13]. In addition, the vaccination of females was identified as a highly cost-effective prevention measure [9,14]. Therefore, despite unpromising statistics, cervical cancer is defined as one of the curable cancers [13].

To decrease the burden of cervical cancer, in 2020, the World Health Organization (WHO) launched a worldwide campaign to eliminate cervical cancer [1,15]. Three major steps mentioned in the WHO’s Global Strategy to Accelerate the Elimination of Cervical Cancer were built around the three pillars of HPV vaccination, screening, and treatment [5,15]. The primary prevention of cervical cancer is based on HPV vaccination; the secondary prevention attempts to identify the disease as early as possible at the premalignant stages, and the tertiary prevention is directed to the management of cervical cancer [5]. Thus, the ‘90–70–90’ cancer elimination targets specify that 90% of teenage girls should receive HPV vaccination before the age of 15, 70% of females should be screened for cervical cancer by the age of 35, and 90% of females diagnosed with cervical neoplastic lesions should receive specific treatment [9,15,16]. Despite having a worldwide scope, this initiative is mostly designed for LMICs. These countries do not have well-established vaccination, screening, and treatment services for cervical cancer [16]. The ‘90–70–90’ target is mostly achievable in high-income countries, where there are sufficient resources to cover the three pillars of the initiative [15,16].

The Republic of Kazakhstan is the largest Central Asian country. The population of the country is 19 million, with females accounting for 52%, of which more than half (51%) are of reproductive age [17,18]. The prevalence of HPV infection in Kazakhstan was unknown [19,20,21] until the recent nationwide study results were published [22,23]. These data show that the prevalence of high-risk HPV strains causing cervical lesions and cervical cancer is high, around 39% among women attending gynecological clinics. Consequently, the cervical cancer incidence in Kazakhstan has been increasing over the past decade [24,25], and cervical cancer is the second leading cancer type among women in the country. Moreover, among Central Asian countries, Kazakhstan has the second highest incidence rate of cervical cancer [24].

HPV vaccination as a primary prevention for cervical cancer and other HPV-related diseases was introduced in Kazakhstan in 2013 [12,26]. Two HPV vaccines, Cervarix and Gardasil, were approved for the campaign [26]. However, the vaccination program was not properly supported by educational and informational interventions [9,12,26]. Moreover, negative content about the vaccination campaign published by local social media has resulted in a negative public reflection, and the program was closed [12,26]. Recently, the Kazakhstani Ministry of Healthcare announced restarting the HPV vaccination program, which is expected in 2024. Although primary prevention is not currently available in the country, a cervical cancer screening program, as an opportunity for secondary prevention, has been launched. It was introduced in Kazakhstan in 2006 and screened women aged 30–34 via the cytological staining method [19,24]. Since then, the screening program has undergone several changes and, as of now, has expanded to include women from 30 to 70 years old who are screened at 4-year intervals with the Papanicolaou test (Pap test) performed by gynecologists in outpatient clinics all over the country. However, the screening coverage was reported to be low, with around 46% of the eligible population [24].

Plans to introduce HPV genotyping as a part of the screening process for cervical cancer was announced in Kazakhstan for 2019–2020; however, such changes are yet to take place, and HPV testing remains an out-of-pocket charge [27]. Despite the efforts to increase the early detection of cervical cancer, the coverage of the free screening program dropped over the period 2008–2016 by 27% [28], and it has had an adverse impact on cervical cancer incidence. Because the natural history of cervical cancer is quite long, the disease could be prevented and cured at the precancerous lesion stage [29,30]. Thus, more attention should be drawn to the epidemiology and management of precancerous cervical lesions. Therefore, the aim of this study is to identify the prevalence of precancerous cervical lesions among non-vaccinated women attending the national tertiary care hospital in Kazakhstan.

## 2. Materials and Methods

### 2.1. Study Sample

This was a retrospective study of the cervical cancer screening database from the national tertiary care hospital—University Medical Center, Republican Diagnostic Center (UMC, RDC), Astana, Kazakhstan. The gynecologic outpatient department of the RDC has around 10,000 visits per year. In this study, the records of all patients who attended the gynecologic outpatient department of RDC in 2018 were reviewed. Potential participants were included in the study if they met the following inclusion criteria: women 18 years and older, who underwent a Pap test in the gynecological outpatient clinic of the RDC in 2018. The RDC gynecological outpatient department attendees who were younger than 18 years old and/or did not have a Pap test in 2018 were excluded. After the database screening, there were 6682 Pap test results identified for the search period (2018). Additionally, available in the RDC electronic database, the clinical records of the patients included in the study were analyzed to collect their demographic and clinical data.

### 2.2. Description of the Laboratory Methods

In the UMC RDC laboratory, Pap test was performed using liquid-based cytology (LBC). Pap test was performed for all the women included in the analysis. The hospital gynecologists followed standard procedures for collecting a cervical sample. A cytobrush was introduced into the external cervical os and rotated to scrape and collect cells from the cervical canal. The cytobrush was then dropped in a vial containing the preservative fluid for LBC, and samples were kept in a container with additive fluid. This fluid removes different types of unwanted debris, such as mucus, blood cells, etc., before setting a layer of cells on the slides. The vial was sent to the RDC cytology laboratory, and the cervical samples were finally placed at a vortex with 3000 rpm for 15–20 s to remove mucosal and blood particles. After adding density reagent to the sample, it underwent sedimentation and centrifugation at 2500× *g* rpm for 5 min. After two alcohol washes, the slides were stained by the Papanicolaou technique according to standard procedure. These slides were then analyzed under the Axioscope 40 and Axiostar plus microscopes (Zeiss, Jena, Germany). Smears were diagnosed and reported using the Bethesda system for cervical cytology [31].

### 2.3. Ethical Approval

The study was performed according to the Helsinki Declaration and was approved by the UMC Institutional Ethical Committee (UMC IRB) in June 2020, decision #4.

### 2.4. Statistical Analysis

All the data were recorded in Excel (Microsoft Office) and analyzed using the statistical software STATA 16 [32]. The data analysis consisted of descriptive statistics in percentages and frequencies. The variables were tested for association by Pearson’s chi-square test and Fischer’s exact test, where appropriate. If the assumption of the expected cell count of more than 5 for the Pearson’s chi-square test was not met, Fischer’s exact test was performed instead. A *p*-value of less than 0.05 was considered statistically significant.

## 3. Results

A total of 6682 women’s records were analyzed in the study. Out of all the 6682 Pap tests performed in 2018, 249 patients (3.73%) had abnormal results and were included in the analysis. The average age of the participants was 40.61 ± 13.62 years. The age distribution of all the participants and their gynecological health conditions are shown in Table 1. According to the available clinical data, none of the patients included in the study received the HPV vaccine in the past.

Out of all the abnormal Pap test results, most of the participants were diagnosed with low-grade squamous intraepithelial lesion (LSIL)—62.25% (Figure 1). The proportion of atypical squamous cells of undetermined significance (ASCUS) and atypical squamous cells, cannot exclude high-grade squamous lesion (ASCs-H) was 18.47%, and high-grade squamous intraepithelial lesion (HSIL) was 19.28%. Thus, the ASCUS + ASCs-H and HSIL results were almost equal among the participants with abnormal cervical cytology findings. Among the women enrolled in the study, 38.15% did not have any comorbidities. The most common coexisting condition was chronic pelvic inflammatory disease (PID) at 24.5%, followed by vaginitis (10.04%), cervical ectropion (5.62%), uterine leiomyoma (5.22%), and infertility (4.82%). The least common coexisting conditions were menopause (3.61%), cervical polyp (2.41%), suspected cervical cancer (2.01%), and other diseases (3.61%). All patients with abnormal test results were managed according to the national guidelines.

The prevalence of squamous lesions stratified by age groups is summarized in Table 2. The highest prevalence of ASCUS and ASCs-H (37%) was observed among older women aged between 38 and 51 years old. The youngest group of women in this study (20–30 years old) had the highest prevalence of LSIL (30%). Women aged between 31 and 37 years old had the highest prevalence of HSIL (33%). As the *p*-value obtained by the Pearson’s chi-square test is greater than the significance value (0.05), the relationship between squamous lesions and age groups is not statistically significant.

The highest prevalence of ASCUS and ASCs-H (46%) was observed among women who had chronic PID and infertility in their past medical history. The highest prevalence of LSIL (41%) and HSIL (27%) in the Pap tests was observed among women who never had any gynecological complications. As the *p*-value obtained by Fisher’s exact test is less than the significance value (0.05), the relationship between squamous lesions and condition groups is statistically significant.

## 4. Discussion

According to recent national healthcare reports, the cervical cancer incidence is growing in Kazakhstan [25]. The absence of a cervical cancer primary prevention program in the form of HPV vaccination makes the accuracy and coverage of cervical cancer screening program in the country of paramount importance. Thus, there is an emergent need to draw more attention to the screening, diagnosis, and management of precancerous cervical lesions. In this study, we aimed to identify the prevalence of premalignant cervical lesions among non-vaccinated women who were referred to the national tertiary care hospital in Kazakhstan.

In this retrospective analysis of the single hospital data, we found that out of all the Pap tests performed in 2018, 3.73% of the cases were abnormal. The prevalence of HSIL was high at 19.28%, and the proportions of ASCUS + ASCs-H and LSIL were 18.47% and 62.25%, respectively. Almost 25% of women had a history of lower and upper genital tract infections, which are well-known to prolong HPV infection persistence, thus, increasing the risk of precancerous cervical lesion development [33]. A small proportion of women who underwent a Pap test due to suspicions of cervical cancer clinical signs, received results with either LSIL or HSIL. Unfortunately, in the cohort of patients we analyzed, the data on their HPV status were not available.

There are limited publications available on the prevalence of precancerous cervical lesions in Kazakhstan. The only study was published by Balmagambetova et al. (2020), which represents regional data from Western Kazakhstan and reported LSIL in 81.6% and HSIL in 2.2% [34]. The rates of LSIL were lower in our study; however, HSIL was identified in almost 10 times the cases than in the compared study [34]. This can be explained by the fact that in our study, we reported the data from the national tertiary care clinic accommodating severe and complicated cases.

To compare our study results, we searched some regional data from the neighboring post-Soviet countries on the same topic. A minimal number of publications are available on the prevalence of precancerous cervical lesions. A study from Russia by Shipitsyna et al. (2011) reported abnormal Pap test results in 9.8% of the cases analyzed [35], which is higher than in our study—3.73%. Most patients in the compared study had ASCUS—73.5%; LSIL was found in 26.5% and HSIL in 1% [35]. There was no information on the concurrent gynecological conditions of the reported patients.

We also compared our results with other LMICs. In a study from Guatemala, the prevalence of abnormal cytology among the general population of women was 7.7% (ASCUS—18%, LSIL—61%, and HSIL—18%) [36], which is comparable with our results. In a study from Cameroon, LSIL was found in 25%, ASCs-H in 14.5%, and HSIL in 3.3% of the cases analyzed [37]. In a study from Turkey, abnormal cytology was observed in 12.9% of the cases: LSIL in 12.4%, HSIL in 3.9%, and ASCUS in 64.3% [38]. These two studies report results that are contrary to our findings. This could be explained by the divergences in population composition (age, ethnicity, education, comorbidities, etc.), the prevalence of HPV infection among the analyzed groups, and HPV vaccination availability.

*Study strength and limitations*. This study has some important strengths and limitations. One of the important strengths of the study is the fact that this is the first analysis of the UMC tertiary care hospital cervical screening database, which will ensure a focus on cervical cancer prevention in the clinic. Moreover, considering the planned relaunching of the HPV vaccination program in Kazakhstan, this study’s results will help to reinforce cervical cancer screening and HPV vaccination promotion. Overall, this study contributes to the scarce local epidemiological data currently available and builds the capacity for future studies. One of the main limitations of this study is the small number of participants and the absence of data on their HPV status and the results of a cervical biopsy. Some more clinical data and patients’ past medical history (contraception, parity, number of children, more data on the history of gynecological diseases, etc.) and the availability of information on social behavioral factors (sexual practices, smoking, environmental, workspace factors, etc.) could refine the quality of the statistical analysis. To improve the accuracy of the statistics and provide more precise nation-wide data, we need to analyze records from the national electronic database including the results of cervical biopsy. This will be a task for future study in this field.

*Clinical implications and future directions*. The awareness of the high rates of abnormal Pap test results that we found in this study will draw specialists’ attention to this women’s health issue. The mindfulness of the healthcare professionals (general practitioners and specialists in general gynecology and gynecologic oncology) about the rates of cervical precancerous lesions could improve the accuracy of screening practices and treatment [39], thus, potentially contributing to a decrease in the cervical cancer rate. Moreover, the awareness of the high prevalence and threat of premalignant cervical lesions will strengthen the focus on the importance of HPV vaccination. These data may serve in the future as a step toward the development of a prediction nomogram estimating the risk of cervical dysplasia recurrence after primary conization and could be used as a tool for counseling women [40]. Further studies on the prevalence of precancerous cervical lesions must include data on high-risk HPV infection distribution and cervical biopsy results.

## 5. Conclusions

Although the overall rate of abnormal cervical cytology results is not significant, the study revealed the high prevalence of HSIL. This fact, together with the absence of primary prevention and the increasing incidence of cervical cancer in the country, calls local policymakers and gynecology specialist to immediate action to improve prophylactic measures: Primary prevention via HPV vaccination has to be restarted and implemented at the national level; the screening coverage must be improved; all women with cervical lesions should be treated according to the guidelines. All these actions will comply with the WHO strategy on cervical cancer elimination and will help to decrease morbidity and mortality from cervical cancer in Kazakhstan. Furthermore, a well-established HPV vaccination program would likely be a very strong measure to reduce not only the incidence and prevalence of precancerous cervical lesions and cervical cancer, but also other HPV-related conditions, such as HPV-associated head and neck cancers.

## Figures and Tables

**Figure 1 healthcare-11-00235-f001:**
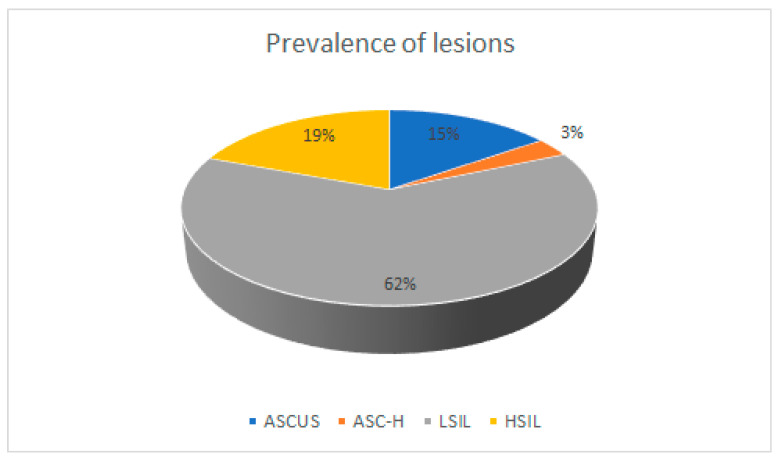
Prevalence of precancerous cervical lesions among the study participants.

**Table 1 healthcare-11-00235-t001:** Patients’ characteristics.

Variables	Patients (N)	Patients (%)
Condition		
Normal	95	38.15
Vaginitis	25	10.04
Cervical ectropion	14	5.62
Cervical polyp	6	2.41
Chronic PID	61	24.5
Infertility	12	4.82
Menopause	9	3.61
Suspected cervical cancer	9	2.01
Uterine leiomyoma	5	5.22
Other	13	3.61
Lesions		
ASCUS + ASCs-H	46	18.47
LSIL	155	62.25
HSIL	48	19.28
Age		
20–30	63	25.3
31–37	67	26.91
38–51	60	24.1
52+	59	23.69

Abbreviations: PID—pelvic inflammatory disease; ASCUS—atypical squamous cells of undetermined significance; ASCs-H—atypical squamous cells, cannot exclude high-grade squamous lesion; LSIL—low-grade squamous intraepithelial lesion; HSIL—high-grade squamous intraepithelial lesion.

**Table 2 healthcare-11-00235-t002:** Squamous lesions stratified by age groups and diseases.

	Categories			*p*-Value
Variables	ASCUS ASCs-H, n (%)	LSIL, n (%)	HSIL, n (%)	
Age				*p* = 0.113 *
20–30	8 (17%)	47 (30%)	8 (17%)	
31–37	12 (26%)	39 (25%)	16 (33%)	
38–51	17 (37%)	30 (20%)	13 (27%)	
52>	9 (20%)	39 (25%)	11 (23%)	
Condition				*p* = 0.002 **
Normal ^1^	18 (39%)	64 (41%)	13 (27%)	
Vaginitis ^2^	1 (2%)	19 (12%)	5 (10%)	
Suspected CC ^2^	0	14 (9%)	11 (23%)	
Chronic PID ^3^	21 (46%)	39 (25%)	13 (27%)	
Other ^4,5^	6 (13%)	19 (22%)	6 (13%)	

* Pearson’s chi-square test; ** Fisher’s exact test, *p* < 0.005; ^1^ women with no gynecological complications; ^2^ women with vaginitis and STI; ^3^ women with suspected cervical cancer and/or diagnosed with cervical ectropion and/or cervical polyp; ^4^ women diagnosed with chronic pelvic inflammatory disease and/or infertility; ^5^ women with one of the following conditions: menopause, uterine leiomyoma, endometrial hyperplasia, gestational trophoblastic disease, PCOS, or endometriosis.

## Data Availability

The study dataset is available from the study PI (balkenzhe.imankulova2@umc.org.kz) per the reasonable request.

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
