# Peer review of "Prevalence of Precancerous Cervical Lesions among Nonvaccinated Kazakhstani Women: The National Tertiary Care Hospital Screening Data (2018)"

_healthcare, 2023, doi:10.3390/healthcare11020235_

Round 1

Reviewer 1 Report

I would like to thanks editors for reviewing invitation, after analyzing the manuscript the topic is interesting although some improvements are require because this subject was already explored by other authors (10.22034/APJCP.2018.19.5.1175) (10.3390/vaccines10010053), I suggest to authors to established some improve on Kazakhstan public or private health system in order to prevent HSIL to worsen.

Author Response

Dear Reviewer,

Thank you very much for the detailed review of our manuscript. We appreciate your time, efforts, valuable comments and suggestion. Please find below our response to your comment.

Comments and Suggestions for Authors

I would like to thanks editors for reviewing invitation, after analyzing the manuscript the topic is interesting although some improvements are require because this subject was already explored by other authors (10.22034/APJCP.2018.19.5.1175) (10.3390/vaccines10010053), I suggest to authors to established some improve on Kazakhstan public or private health system in order to prevent HSIL to worsen.

Response – Thank you for the comment. We agree that the subject of the manuscript is interesting. We have explored a lot on it in our country and the articles you suggested here (10.22034/APJCP.2018.19.5.1175; 10.3390/vaccines10010053) that has been already published by our team are an evidence of our dedication to the research on HPV and cervical cancer in Kazakhstan. However, these articles are presented in a form of review, while our current submission that you kindly agreed to review is a research manuscript, which resulted from our recent retrospective study. The article aims to focus attention on precancerous cervical lesions that should help to improve the management. Moreover, as Kazakhstani Ministry of Healthcare is going to re-launch the HPV vaccination program in 2024, this manuscript will increase focus and understanding on the vaccination issues in the country as well.

Dear Reviewer, per your suggestions, some changes incorporated into the manuscript text to improve its quality. Please see the resubmitted version with track changes.

Reviewer 2 Report

I read with great interest the Manuscript titled "Prevalence of precancerous cervical lesions among non-vaccinated Kazakhstani women: the national tertiary care hospital screening data (2018)." which falls within the aim of the Journal.

In my honest opinion, the topic is interesting enough to attract the readers’ attention. 

Nevertheless, authors should clarify some point and improve the discussion citing relevant and novel key articles about the topic.

-The introduction should be extended and completed. I find interesting a reference to the efforts made for the prevention and early diagnosis of gynecological cancers (see PMID: 36141217).

- Although it is a retrospective analysis, inclusion/exclusion criteria should be better clarified by extending their description.

- The authors have not adequately highlighted the strengths and limitations of their study. I suggest better specifying these points. The sentence in line 216 should be revised.

- I suggest also authors to not repeat some concepts more than once and to organize better the discussion section following this ideal structure: main findings of the study, strength and limitations of the study, implications and comparison with literature, future directions. (i.e., lines 180-182, 216-217 and 231-232 are almost the same).

- Discussions can be expanded and improved by citing relevant articles (I suggest authors to read and insert in references the following article PMID: 35909005; 35455328).

Considered all this points, I think it could be of interest for the readers and, in my opinion, it deserves the priority to be published after minor revisions.

Author Response

Dear Reviewer,

Thank you very much for the detailed review of our manuscript. We appreciate your time, efforts, valuable comments and suggestions that helped us to improve the text quality. Please find below our point-by-point responses for all your comments.

Comments and Suggestions for Authors

I read with great interest the Manuscript titled "Prevalence of precancerous cervical lesions among non-vaccinated Kazakhstani women: the national tertiary care hospital screening data (2018)." which falls within the aim of the Journal.

In my honest opinion, the topic is interesting enough to attract the readers’ attention. 

 Nevertheless, authors should clarify some point and improve the discussion citing relevant and novel key articles about the topic.

 -The introduction should be extended and completed. I find interesting a reference to the efforts made for the prevention and early diagnosis of gynecological cancers (see PMID: 36141217).

 Response – Dear Reviewer, thank you for the comment. The introduction part has been reworked and the suggested article and some more other citations are included into the reference list. Please see the resubmitted manuscript version with highlighted changes.

- Although it is a retrospective analysis, inclusion/exclusion criteria should be better clarified by extending their description.

Response – Thank you for the comment. Relevant amendments were performed in the text of the Methods section.

“Potential participants were included into the study if they met the following inclusion criteria: women of 18 years and older, who undergone Pap-testing in the gynecological outpatient clinic of the RDC in 2018. The RDC gynecological outpatient department attendees who are younger than 18 years old and/or have not passed Pap-test in 2018 were excluded. After the database screening, there were 6,682 Pap-tests results identified for the searching period (2018). Additionally, available in the RDC electronic database clinical records of the patients included into the study were analyzed to collect their demographic and clinical data.”

- The authors have not adequately highlighted the strengths and limitations of their study. I suggest better specifying these points. The sentence in line 216 should be revised.

Response – Thank you for the comment. The study strength and limitations text was revised The sentence in Line 216 is rephrased.

- I suggest also authors to not repeat some concepts more than once and to organize better the discussion section following this ideal structure: main findings of the study, strength and limitations of the study, implications and comparison with literature, future directions. (i.e., lines 180-182, 216-217 and 231-232 are almost the same).

Response – Thank you for the comment. The discussion part was reworked. The study rational, main findings, comparison with literature, strength and limitations, possible implications, and future directions are mentioned. he pointed repeated sentences were removed/rewritten. Please see the resubmitted text with highlighted changes.

- Discussions can be expanded and improved by citing relevant articles (I suggest authors to read and insert in references the following article PMID: 35909005; 35455328).

Response - Thank you for the comment. The discussion part was reworked and the suggested references are cited.

Considered all this points, I think it could be of interest for the readers and, in my opinion, it deserves the priority to be published after minor revisions.

Response – thank you very much for your kind consideration of our work and the revision that helped to improve the text.

Reviewer 3 Report

The authors Imankulova et al. conducted a retrospective study on the prevalence of pre-cancerous cervical lesions in Kazakhstan. Test results and clinical data of more than 6,000 tests were screened and analyzed. The authors found a comparably high prevalence in their study population which is comparable to other studies performed in low and middle income countries.

Overall, the study seems to be conducted well and the results as well as the limitations of the study are well discussed. English language is fine. As an international reviewer, I am not familiar with the local health policies, especially regarding an HPV vaccination program and would emphasize the authors to add some information to the history to the local vaccination program being suspended. As it should also be discussed, a good vaccination program would likely be a very strong measure to reduce the number of precancerous cervical lesion and cervical cancer as well as other entities such as HPV associated head and neck cancer.

Author Response

Dear Reviewer,

Thank you very much for the detailed review of our manuscript. We appreciate your time, efforts, valuable comments and suggestions that helped us to improve the text quality. Please find below our point-by-point responses for all your comments.

Comments and Suggestions for Authors

The authors Imankulova et al. conducted a retrospective study on the prevalence of pre-cancerous cervical lesions in Kazakhstan. Test results and clinical data of more than 6,000 tests were screened and analyzed. The authors found a comparably high prevalence in their study population which is comparable to other studies performed in low and middle income countries.

Overall, the study seems to be conducted well and the results as well as the limitations of the study are well discussed. English language is fine. As an international reviewer, I am not familiar with the local health policies, especially regarding an HPV vaccination program and would emphasize the authors to add some information to the history to the local vaccination program being suspended. As it should also be discussed, a good vaccination program would likely be a very strong measure to reduce the number of precancerous cervical lesion and cervical cancer as well as other entities such as HPV associated head and neck cancer.

Response - Thank you very much for your kind consideration of our work and the revision that helped to improve the text. As was suggested some historical context related to the HPV vaccination in Kazakhstan were included into the introduction part.

“HPV vaccination as a primary prevention for cervical cancer and other HPV-related diseases was introduced in Kazakhstan in 2013 [17,20]. Two HPV vaccines, Cervarix and Gardasil, were approved for the campaign [20]. However, the vaccination program was not properly supported by educational and informational intervention [17,20, Akhatova 2022]. Moreover, a negative content published by the local social media have resulted with a negative public reflection and the program was closed. Recently the Kazakhstani Ministry of Healthcare has announced restarting the HPV vaccination program, which is expected in 2024.”

As it should also be discussed, a good vaccination program would likely be a very strong measure to reduce the number of precancerous cervical lesion and cervical cancer as well as other entities such as HPV associated head and neck cancer.

Response - Thank you very much for your kind consideration of our work and the revision that helped to improve the text. As was suggested possible impact of HPV vaccination is described in the discussion  and conclusion part.

“Although the overall rate of abnormal cervical cytology results is not significant, the study revealed the high prevalence of HSIL. This fact, together with the absence of primary prevention and increasing incidence of cervical cancer in the country, is calling the local policymakers and gynecology specialist for immediate action to improve the prophylactic measures: primary prevention via HPV vaccination have to be restarted, and implemented at the national level; the screening coverage must be improved; all women with cervical lesions should be treated according to the guidelines. All these actions will be in compliance with the WHO strategy on cervical cancer elimination and will help to decrease morbidity and mortality from cervical cancer in Kazakhstan. Furthermore, a well-established HPV vaccination program would likely be a very strong measure to reduce not only the incidence and prevalence of precancerous cervical lesions and cervical cancer, but also other HPV-related conditions such as HPV-associated head and neck cancer.”

Round 2

Reviewer 1 Report

No comments

Author Response

Dear Reviewer,   Thank you very much for reviewing our manuscript and making valuable and helpful comments. We have revised the Introduction,  results, and discussion part.   Reviewer Comments and Suggestions for Authors

No comments.